# Population Genetics of the Blueberry Gall Midge, *Dasineura oxycoccana* (Diptera: Cecidomyiidae), on Blueberry and Cranberry and Testing Invasion Scenarios

**DOI:** 10.3390/insects13100880

**Published:** 2022-09-28

**Authors:** Hyojoong Kim, Cesar Rodriguez-Saona, Heung-Sik Lee

**Affiliations:** 1Animal Systematics Laboratory, Department of Biological Science, Kunsan National University, Gunsan 54150, Korea; 2Department of Entomology, P.E. Marucci Center, Rutgers University, Chatsworth, NJ 08019, USA; 3Animal & Plant Quarantine Agency, Gimcheon 39660, Korea

**Keywords:** Cecidomyiidae, cranberry tipworm, exotic pests, host-associated differentiation, invasion route, molecular ecology

## Abstract

**Simple Summary:**

The blueberry gall midge—*Dasineura oxycoccana* (Johnson) (Diptera: Cecidomyiidae)—is an economically important insect pest of blueberry and cranberry in its native range (USA and Canada) as well as in other parts of the world. This pest was recently introduced and spread through other regions within North America as well as in Europe and Asia, e.g., Korea. To confirm that incipient speciation might have occurred in *D. oxycoccana* populations associated with blueberry and cranberry as previously reported, it is necessary that approaches on population genetics are performed with a larger sample size. To identify possible routes of worldwide introductions, it is important to trace the source and invasion process of the Korean population. Therefore, we compared the population genetic structure between *D. oxycoccana* populations from blueberry and cranberry from USA and Korea. We found (1) a clear separation between the two host-associated *D. oxycoccana* populations from blueberry and cranberry, which could be considered distinct species; (2) the occurrence of five genetically isolated *D. oxycoccana* subgroups from blueberry; (3) that multiple *D. oxycoccana* introductions likely occurred in Korea; and (4) that the dominant invasive *D. oxycoccana* population from Korea was likely introduced from a genotypically diverse population, which was likely introduced from an unsampled source population rather than directly from its native range.

**Abstract:**

We compared the population genetic structure between populations of the blueberry gall midge—*Dasineura oxycoccana* (Johnson) (Diptera: Cecidomyiidae)—from blueberry and cranberry and determined the genetic relationships among geographical subgroups by genotyping 632 individuals from 31 different populations from their native USA regions (New Jersey, Michigan, and Georgia) and from invaded Korean regions using 12 microsatellite loci. Our population genetic analyses showed a clear separation between the two host-associated *D. oxycoccana* populations from blueberry and cranberry. Using data from only the blueberry-associated *D. oxycoccana* populations, we identified five genetically isolated subgroups. An analysis of the approximate Bayesian computation suggests that the invasive *D. oxycoccana* population from Korea appears to have been introduced from an unsampled source population rather than directly from its native range. Our findings will allow for an easier identification of the source of *D. oxycoccana* into newly invaded regions, as well as to determine their association with blueberry and cranberry, which based on our results can be considered as two distinct species.

## 1. Introduction

The blueberry gall midge—*Dasineura oxycoccana* (Johnson) (Diptera: Cecidomyiidae)—is native to central and eastern North America (United States of America and Canada), where the ancestors of cultivated blueberry species, such as highbush blueberry (*Vaccinium corymbosum* L.), lowbush blueberry (*Vaccinium angustifolium* Aiton), and rabbiteye blueberry (*Vaccinium virgatum* Aiton), grow in the wild [1,2,3,4,5]. In 2004, *D. oxycoccana* was found in eastern US states (Oregon and Washington); it likely transferred from infested blueberry nursery plants that originated from other blueberry-producing states [6]. In Canada, *D. oxycoccana* is found in British Columbia, New Brunswick, and Nova Scotia [7], although this species is considered only a secondary pest in northeastern Canada [8]. Like in the United States, the movement of this pest within Canada likely occurred through the commercial trade of blueberry nursery plants [2,9]. This insect is an economically important pest in its native range [5], affecting most cultivated blueberry (*Vaccinium* spp.) species, including rabbiteye and highbush blueberries, and causing substantial yield loss by injuring flower and leaf buds [10]. In North America, *D. oxycoccana* larval feeding can kill up to 80–90% of the flower buds [1,11]. In addition, *D. oxycoccana* injury to leaf buds can result in leaf curl, stunted growth, and blackened leaf tips [12,13]. In particular, *D. oxycoccana* is a major pest in southern US states, such as Florida, where they can injure the flower buds because they are active earlier in the growing season than in northern states [1]. As a result, the annual economic losses by *D. oxycoccana* to the Florida blueberry industry can be up to US 20 million [14].

In North America, in addition to blueberries, this pest also feeds on cranberries (*Vaccinium macrocarpon* Aiton) and can be found in most cranberry-growing regions such as Massachusetts and Wisconsin [5], where it is referred to as the cranberry tipworm [15,16]. Although *D. oxycoccana* populations from cranberry are morphologically indistinguishable from those from blueberry, they are thought to be two distinct species based on their mating behavior, genetics, and pheromone composition [9,16,17]. These previous studies indicate that *D. oxycoccana* populations from blueberry and cranberry are likely genetically isolated and have likely developed host specificity due to their use of two different host plant species, resulting in incipient speciation [9,16,17]. Nevertheless, the existence of host-associated differentiation (HAD) between *D. oxycoccana* populations from blueberry and cranberry has not yet been confirmed based on an extended genetic analysis.

Due to growth in the global trade and production of blueberries [18], *D. oxycoccana* is becoming a threat to blueberries across the world. Within North America, it was recently reported in Mexico [19]. Outside of North America, it was first introduced into Europe around 1996 [7], with the first European invasion confirmed in Northern Italy, although the blueberry nursery plants carrying *D. oxycoccana* originated from Germany [20,21]. In Europe, *D. oxycoccana* is currently distributed in The Czech Republic, Germany, France, Italy, Latvia, Lithuania, The Netherlands, Poland, and Romania [22]. Its distribution is limited to some regions in the United Kingdom, but it appears widespread in England [7]. In Asia, *D. oxycoccana* was confirmed in Japan in 2015 [23], while no official record has been reported so far in China. In Korea, *D. oxycoccana* was found about a decade ago causing injury to blueberries [24,25]. In recent years, as blueberry cultivation in Korea expanded rapidly from 30 ha in 2005 to 3369 ha in 2020 [24,26], *D. oxycoccana* became a serious pest of this crop [10,24]. Since its initial invasion, *D. oxycoccana* has rapidly spread throughout Korea [10,24]. Because this pest is not considered a migratory species that can move through long distances for reproduction [27], *D. oxycoccana* populations are expected to be isolated and fragmented regionally in cultivated blueberry farms. Still, little is known about the population genetic structure of this pest in native and invaded regions.

Tracing the source of an invasive pest and gaining a better understanding of its population genetic structure are important to prevent continuous and multiple introductions and to develop control strategies such as the importation of the pest’s natural enemies [28,29,30]. For instance, several *D. oxycoccana* host races are expected to occur in Korea due to the diverse global trade routes of imported blueberry plants [31], yet the origin and number of host races of *D. oxycoccana* in this country remain unknown. Although *D. oxycoccana* is native to central and eastern North America, it has been in Europe for over two decades, and most blueberry nursery plants imported into Korea come from China and Japan [31]; thus, *D. oxycoccana* populations from Korea could have originated from various sources. 

The main objectives of this study were to compare the population genetic structure of *D. oxycoccana* between (1) populations collected from cranberries and blueberries and (2) a region of origin—USA—and an invaded region—Korea. Specifically, we first compared *D. oxycoccana* samples collected from blueberries and cranberries to test for HAD in this species. Second, we characterized the genetic diversity of *D. oxycoccana* populations from USA and Korea and tested for genetic differentiation among the regional populations. For these studies, *D. oxycoccana* samples were collected at the beginning of the Korean invasion and were analyzed using 12 microsatellite loci that were previously developed [25]. Polymorphic microsatellite loci are an effective tool for studying fundamental questions regarding the population genetics of invasive pests [32,33,34]. Finally, we inferred the most suspected source of the *D. oxycoccana* invasive population from Korea by using the standard analysis—approximate Bayesian computation (ABC). 

## 2. Materials and Methods

### 2.1. Collection Sites

Due to the fact that none of the *D. oxycoccana* sample collections in this study were carried out in restricted areas—national parks, etc.—where permits are required, there was no need for special collection permits. We examined a total of 632 *D. oxycoccana* individuals obtained in 2011–2013 from 31 different population collections from Korea (22 collections) and USA (9 collections) (Figure 1; Appendix A). Of these collections, 28 were obtained from highbush, lowbush, and rabbiteye blueberries (*Vaccinium* spp.) in Korea and USA, while three collections were obtained from cranberries (*V. macrocarpon*) in USA. Samples from US populations were collected from the central (Michigan), southern (Georgia), and eastern (New Jersey) regions, which comprise the native range of *D. oxycoccana* and possible sources of invasion into Korea. In the invaded region, samples from Korean populations were collected from areas where *D. oxycoccana* occurred during the period of initial invasion (2011–2013). All samples consisted of *D. oxycoccana* larvae, which is the damaging stage living inside the plant tissues, and thus, they have a strong association with their host plant. To avoid sampling related (sibling) *D. oxycoccana* individuals, we collected specimens for the molecular analyses from different host plants that were distantly located. All freshly (live) collected *D. oxycoccana* larvae used for molecular analyses were carefully removed from infested buds and preserved in 95% or 99% ethanol and stored at −70 °C.

### 2.2. Microsatellite Genotyping

A total of 632 *D. oxycoccana* individuals were genotyped using 12 microsatellite loci (Dox08, Dox09, Dox10, Dox11, Dox12, Dox22, Dox23, Dox25, Dox30, Dox33, Dox41, and Dox42), which were previously isolated from this species [25]. In a preliminary test, all loci developed in the previous study [25] were polymorphic among most population samples and were thus included in the analyses. Total genomic DNA was extracted from single individuals using the LaboPass™ Tissue Genomic DNA Mini Kit (COSMOGENETECH, Daejeon, Korea) according to the manual’s protocol. All genomic DNA templates were extracted from the whole body of *D. oxycoccana* larvae, which were mostly at the final instar stage. The tissues were left in the lysis buffer with protease *K* solution at 55 °C for 24 h, and then the cleared cuticle was dehydrated. Microsatellite amplifications were performed using AccuPower^®^ PCR PreMix K-2037 (BIONEER, Daejeon, Korea) in 20-μL reaction mixtures containing 0.5 μM forward primer labeled with a fluorescent dye (6-FAM, HEX, or TAMRA), reverse primers, and 0.05 μg of the DNA template. Polymerase chain reaction (PCR) was performed using a GS482 thermo-cycler (Gene Technologies, Essex, UK) according to the following procedure: initial denaturation at 95 °C for 5 min, followed by 34 cycles of 95 °C for 30 s, annealing at 56 °C for 40 s, extension at 72 °C for 45 s, and a final extension at 72 °C for 5 min. PCR products were visualized by electrophoresis on a 1.5% agarose gel with a low range DNA ladder to check for positive amplifications. Automated fluorescent fragment analyses were performed on the ABI PRISM 377 Genetic Analyzer (Applied Biosystems, Waltham, MA, USA), and allele sizes of PCR products were calibrated using the molecular size marker—ROX labeled-size standard (GenScan^TM^ ROX 500, ABI, Waltham, MA, USA). Raw data on each fluorescent DNA products were analyzed using GeneMapper^®^ version 4.0 (ABI, Waltham, MA, USA).

### 2.3. Data Analyses

For the 632 individual samples, the results of allele data analyses were processed in GENALEX 6.503 [35] through Microsoft office Excel 2019 (Microsoft). We used GENCLONE 2.0 [36] to identify multilocus genotypes (MLGs) among *D. oxycoccana* populations [37]. We estimated observed (*H*_O_) and expected heterozygosity (*H*_E_) values on multiple loci among the population datasets and between the regional datasets using GENEPOP 4.0.7 [38]. By using sequential Bonferroni correction for all tests involving multiple comparisons [39], we calculated linkage disequilibrium and levels of significance for Hardy–Weinberg equilibrium (HWE). For heterozygote excess or deficiency, values deviated from HWE were estimated. We used MICRO-CHECKER 2.2.3 [40] to identify possible scoring errors due to the large allele dropout as well as stuttering and null alleles [41]. To calculate gene diversity (*H*_S_), mean number of alleles (*N*_A_), allelic richness (*R*_S_), and inbreeding coefficient (*F*_IS_), we used FSTAT 2.93 [42]. 

Different groupings were tested independently on the (1) ecological basis (cranberry associated versus blueberry associated; ‘Case 1′), (2) geographical basis (source versus invasion; *D. oxycoccana* from blueberry only; ‘Case 2′), and (3) genetic structure-based groups (A, B, C, D, E; *D. oxycoccana* from blueberry only; ‘Case 3′) with analysis of molecular variance (AMOVA) in ARLEQUIN 3.5.1.2 [43], with significance determined using the non-parametric permutation approach described by Excoffier et al. [44]. We also used ARLEQUIN for the calculations of pairwise genetic differentiation (*F*_ST_) values [45], in which 31 populations were assigned by each local collection. An exact test of population differentiation was done as optioned by 100,000 Markov chains, 10,000 dememorization steps, and a 0.05 significance level.

The program BOTTLENECK 1.2.02 [46] was used to identify the possible effect of a recent bottleneck in our samples, separately for each population. Two mutation models—considered appropriate for microsatellites [46,47]—were applied as the strictly stepwise mutational model (SMM) and the two-phase model (TPM). For the TPM, a model that includes both 90% SMM and 10% TPM was used for 20,000 iterations. Significant deviations in observed heterozygosity over all loci were tested using a nonparametric Wilcoxon signed-rank test [46,47].

To test for genetic associations among the 31 *D. oxycoccana* populations, principal coordinate analysis (PCoA) was used by calculating the genetic distance matrix with codominant genotypic distance [48,49] implemented in GENALEX 6.503 [35]. The PCoA allows for the visualization of major patterns based on a multilocus genotype with multiple samples [48,49]. Plots in the PCoA were independently calculated and displayed by either individual- or population-level distances.

We used STRUCTURE 2.3.4 [50] to obtain the genetic structure of the 31 *D. oxycoccana* populations by the Bayesian clustering algorithm. The number of clusters (*K*) from 1–11 was set and conducted in 10 independent replications for each *K* value. In each run, after a burn-in period of 30,000 steps, 500,000 Markov chain Monte Carlo repetitions were performed with an allowed admixture model. We obtained the Δ*K* value by calculating ‘∆*K* = m(|*L*′′(*K*)|)/s[*L*(*K*)]’ and applying the ad hoc quantity based on the second-order rate of the likelihood change [51]. Using the STRUCTURE HARVESTER 0.6.94 [52]—which analyzes the data structure—we calculated ∆*K* correctly. The results of STRUCTURE were visualized by conducting DISTRUCT 1.1 [53]. 

We used GENECLASS2 [54] to estimate the assignment/exclusion probabilities for the detection of genetic signatures of dispersal/immigration. For each individual belonging to a population, the program estimated the likelihood values of any other reference population (not home) or the population in which it was collected (home). The sample showing the highest assignment probability was determined to be the most likely source of the assigned genotype. A Bayesian method for estimating an allele frequency of the population [55] was used to test the significance of assignments (type I error, alpha = 0.01) [56] with Monte Carlo resampling computation of 10,000 simulated individuals.

### 2.4. ABC Analysis

To estimate the relative likelihood of alternative scenarios of the *D. oxycoccana* invasion, an ABC analysis was performed for microsatellite data, as implemented in DIYABC 2.1.0 [57]. The genetic admixture events in introduced populations—serial or independent introductions—and the comparison of complex scenarios involving bottlenecks can be estimated in DIYABC [58]. The parameters for simulating scenarios were the times of split or admixture events, the duration of the bottleneck during colonization, the stable effective population size, the rate of admixture, and the effective number of settlers in the invasive populations [59]. This produces a simulated dataset used to calculate the posterior distribution of parameters to choose the most likely scenario [59]. DIYABC makes a simulated dataset for selecting the one most similar to the observed dataset (*n_δ_*) that is ultimately applied to estimate the posterior distribution of the parameters [60].

The DIYABC analysis was conducted for the purpose of inferring (1) the serial divergence between blueberry and cranberry host races in Analysis #1 and (2) the initial introduction process of *D. oxycoccana* from the source (North American) to the invaded (Korean) regions in Analysis #2. Considering the results of PCoA, STRUCTURE, and GENECLASS2 (see Results section), several populations could be selectively used to estimate the scenarios based on their relationships. In the first ABC analysis that tested the divergence between blueberry and cranberry *D. oxycoccana* populations—Analysis #1—we set one cranberry-associated group (CR) with populations US-C-NJ4, US-C-MA, and US-C-WC and three blueberry-associated groups, namely the New Jersey (USA) group (BBNJ) with populations US-B-NJ1, US-B-NJ2, and US-B-NJ3; the Georgia (USA) group (BBGA) with populations US-B-GA1 and US-B-GA2; and the Michigan (USA) group (BBMG) with population US-B-MG. Three scenarios (1–3) were estimated with comparison to each other in the DIYABC (Appendix A). In the second ABC analysis (Analysis #2) that tested the introduction process of *D. oxycoccana* from the source (USA) to the invaded region (Korea), we set one source blueberry-associated group (SCNJ) with populations US-B-NJ1, US-B-NJ2, and US-B-NJ3 and two invasive blueberry-associated groups, namely one invasive group A (INVA) with populations KR-B-UW, KR-B-HS1, KR-B-KY, KR-B-CW, KR-B-YD, KR-B-DA, KR-B-IS, KR-B-DJ, KR-B-SC, KR-B-HW, KR-B-NH, and KR-B-JJ1 and another invasive group B (INVB) with populations KR-B-GJ, KR-B-HS2, KR-B-PT, KR-B-YS, KR-B-SJ, and KR-B-BH1. SCNJ was set as the source group because all populations within this group came from New Jersey (USA), where *D. oxycoccana* originates and is thus a likely source of invasion into Korea. Three scenarios (1–3) were also estimated with comparison to each other in the DIYABC (Appendix A).

We produced 1,000,000 simulated datasets for each scenario. We used a generalized stepwise model (GSM) as the mutational model for microsatellites, which assumes increases or reductions by single repeat units [60]. To identify the posterior probability of these three scenarios, the *n_δ_* = 30,000 (1%) simulated datasets closest to the pseudo-observed dataset were selected for the logistic regression, which were similar to the *n_δ_* = 300 (0.01%) ones for the direct approach [59]. The summary of statistics was calculated from the simulated and observed data for each of the tested scenarios, such as the mean number of alleles per locus (*A*), mean genetic diversity for each group and between group, genetic differentiation between pairwise groups (*F_ST_*), classification index, shared alleles distance (*D_AS_*), and Goldstein distance.

## 3. Results

### 3.1. Population Genetics Analyses

In this study, we genotyped 632 *D. oxycoccana* samples using 12 microsatellite loci, which were all discovered to be non-clonal MLGs—i.e., non-identical genotypes estimated by multiple loci (Table 1). Therefore, in all groups, the number of MLGs was the same as those of individuals in each population. The observed (*H*_o_) and expected (*H*_E_) heterozygosity values from all 31 populations were in the ranges of 0.617–0.898 (averaging 0.764) and 0.538–0.789 (averaging 0.705), respectively (Table 1). According to HWE, there were significant deviations in the KR-B-HE, KR-B-DJ, and KR-B-NH populations by heterozygote excess (Table 1), which was affected by heterosis or over-dominance related to selection preference toward a heterozygous combination or fixation of heterozygous genotypes. However, there were significant deviations in the KR-B-JJ1 and US-B-MG populations by heterozygote deficit (Table 1), which was affected by retaining numerous unique genotypes with private alleles within a population related to their relatively high *H*_E_ [61]. Gene diversity (*H*_S_), the mean number of alleles (*N*_A_), and allelic richness (*R*_S_) averaged 0.70, 6.44, and 1.70, respectively, and the inbreeding coefficient (*F_IS_*) was mostly a negative value (−0.10 ± 0.13, mean ± SD). Generally, positive *F_IS_* values indicate that some number of heterozygous offspring in the population decreased, usually due to inbreeding, whereas negative *F_IS_* values indicate an increase in heterozygosity due to random mating or outbreeding [62].

We estimated pairwise genetic differentiation (*F*_ST_) among the 31 blueberry and cranberry *D. oxycoccana* populations (Appendix A). Except for eight non-significant values, the pairwise comparisons of the *F*_ST_ values showed that the blueberry populations were largely different genetically from the cranberry populations, of which the mean *F*_ST_ between them was 0.276. The mean *F*_ST_ within the blueberry populations was 0.120, whereas the mean *F*_ST_ within the cranberry populations was 0.106. The *F*_ST_ values among the Korean *D. oxycoccana* populations averaged 0.106, whereas the *F*_ST_ values among the USA populations averaged 0.143. Eight pairwise *F*_ST_ values, such as KR-B-PT versus KR-B-GJ, were not significant and had very low or negative values (−0.033 to 0.080), which indicates that, genetically, they are very similar to each other. 

To confirm the molecular variance among the preordained groups, three cases were tested using AMOVA implemented in ARLEQUIN [43,44]. The genetic variance among groups in Case 1 was 14.58%, which suggests that there are relatively large differences between the blueberry and cranberry *D. oxycoccana* populations (Table 2). Excluding cranberry populations, the genetic variance among groups in Case 2 was smaller (7.49%) than that in Case 3 (10.57%), whereas the genetic variance among populations within groups in Case 2 (8.34%) was larger than that in Case 3 (4.32%). Therefore, STRUCTURE-based groups have more genetic differences among the preordained groups within all blueberry *D. oxycoccana* populations than the opposite case. These results support the notion that some invasive groups in Korea are genetically close to the native groups in the USA, regardless of geographic distance. 

Based on the results from BOTTLENECK [46], a significant observed heterozygosity excess (*p* < 0.05, one tail) from the Wilcoxon sign-rank tests (both SMM and TPM) was detected in only two *D. oxycoccana* populations—KR-B-HS1 and KR-B-HW—and the shifted mode was observed in the four populations—KR-B-DJ, KR-B-HW, US-B-GA1, and US-B-NJ1 (Appendix A). Among them, the population KR-B-HW seemingly underwent genetic bottleneck because it was significant in all analyses; nonetheless, the bottleneck test should be interpreted cautiously because the sample size for some populations was less than 30 individuals [47].

PCoA was used to create three independent plots of the datasets with and without the cranberry populations. The first PCoA plot based on individual distances included the cranberry populations and shows that the blueberry populations—located mostly in three quadrants—and the cranberry populations—located mostly in one quadrant—were perfectly separated from each other (Figure 2). The second PCoA plot, which included the cranberry populations but was based on population distances, also shows that the blueberry populations were largely different from the cranberry ones based on the codominant-genotypic distance of multilocus (Appendix A). Our third PCoA plot excluded the cranberry populations from the dataset and showed the blueberry populations were structured into five genetically isolated subgroups, which are labeled as subgroups A, B, C, D, and E (Figure 3).

In all STRUCTURE analyses, the most likely number of clusters was estimated using the Δ*K* calculation based on the Evanno et al. [51] method (Appendix A). We found that the best value was 985.98 on Δ*K =* 2 and the second-best value was 64.23 on Δ*K =* 6 after testing from *K* = 1 to *K* = 10 (Appendix A). Although *K =* 2 provided the best estimate, the results from *K =* 3 to *K =* 6 were further considered because they most effectively displayed the relationships among *D. oxycoccana* populations when compared with other analyses such as PCoA (Figure 3). The STRUCTURE result of *K =* 2 for all samples resulted in two clusters (green and red), showing that some of the blueberry populations fall into the red together with the cranberry populations or had mixed assignments of green and red (Figure 4). Interestingly, the red cluster was almost completely converted to blue in the result of *K* = 3, except for the cranberry populations (Figure 4). A light-blue cluster occurred for some US and Korean *D. oxycoccana* populations in the result of *K* = 4, and then, a yellow cluster occurred for some Korean populations in the result of *K* = 5 (Figure 4). Finally, the STRUCTURE result of *K =* 6 showed five distinct clusters for the *D. oxycoccana* populations, partitioned into five subgroups (A, B, C, D, and E) (Figure 4), which are consistent with those from the PCoA (Figure 3).

Based on the assignment test using GENECLASS2 (Appendix A), which shows the average probability with which samples were destined to the most likely reference population, the values of self-assignment probability in *D. oxycoccana* were 0.397 ± 0.132 (mean ± SD) for the blueberry populations, 0.526 ± 0.047 for the cranberry populations, 0.328 ± 0.140 in the US populations, and 0.415 ± 0.126 for the Korean populations. The highest values of non-self-assignment probability from residence to the expected source between *D. oxycoccana* populations were detected as being most likely in the first KR-B-UW (from five populations), the second KR-B-GJ (from four populations), and the third US-B-MG (from three populations).

### 3.2. Inferring Introductions from Source to Invaded Regions

Most of the 31 *D. oxycoccana* populations were assigned to three or four subgroups according to the results from the PCoA and STRUCTURE (Figure 3 and Figure 4; Appendix A), in which each of the subgroups formed a larger group containing several populations with a similar genetic structure. In addition, we included an unsampled subgroup in the scenarios because one or more introductions could occur from an undetected (=unsampled) population [63].

Analysis #1 tested for the simulated comparison of three scenarios to infer the serial divergence between populations from blueberry and cranberry in the native range (Figure 5; Appendix A) and showed that the BBNJ (New Jersey blueberry), BBGA (Georgia blueberry), and BBMG (Michigan blueberry) groups swap positions with each other in a same clade, which indicates that one at the most basal position diverged from the CR (cranberry-associated) group and then the two remaining subgroups diverged from the unsampled group. As a result of Analysis #1, a Scenario 2 was estimated to be the most likely of the three scenarios, showing a posterior probability ranging from 0.475 (*n_δ_* = 3000) to 0.517 (*n_δ_* = 30,000) with a 95% CI of 0.404–0.547 and 0.491–0.543, respectively, which predicts that BBGA first diverged from CR and then BBNJ and BBMG later diverged from an unsampled group (Figure 5).

Analysis #2 tested for the simulated comparison of the three scenarios to infer the initial introduction process of *D. oxycoccana* from a source (USA) into an invaded (Korea) region (Figure 5; Appendix A). Scenario 1 predicted that INVA (invasive group A) and INVB (invasive group B) subgroups serially branched off from the unsampled group, which diverged from the SCNJ (source) group. Scenario 2 predicted that INVA first diverged from the unsampled group and that later the SCNJ and INVB groups serially branched off. Scenario 3 predicted that the SCNJ and INVA groups first diverged from an unsampled group and that later the INVB group arose from an admixture event of both the SCNJ and INVA groups. As a result of Analysis #2, Scenario 1 was estimated to be the most likely of the three, showing a posterior probability ranging from 0.901 (*n_δ_* = 3000) to 0.881 (*n_δ_* = 30,000) with a 95% CI of 0.886–0.927 and 0.873–0.889, respectively (Figure 5).

## 4. Discussion

### 4.1. Ecological Speciation between the Two Host Races in Blueberry and Cranberry

Regarding possible ecological speciation between the two races of *D. oxycoccana* from blueberry and cranberry, we found that the two subgroups consisting of different host-associated populations collected from blueberry and cranberry were clearly separated by our population genetic analyses—PCoA and STRUCTURE (Figure 2 and Figure 4; Appendix A). These results—which included a large sample size—strongly corroborate previous studies that show ecological speciation between cranberry-associated and blueberry-associated *D. oxycoccana* populations [9,16,17]. In an earlier study, Cook et al. [17] found that *D. oxycoccana* individuals from cranberry and blueberry hosts display complete assortative mating, showing the potential for host race formation or cryptic speciation. In British Columbia, highbush blueberry (*V. corymbosum*) plants bloom several weeks before cranberry (*V. macrocarpon*) plants [17], which could lead to different *D. oxycoccana* life cycles and/or behaviors—such as differentiation in phenology, courtship, and pupation sites—resulting in assortative mating among populations from these host plants [17]. Due to these ecological barriers, *D. oxycoccana* has likely diverged into two distinct species that specialize on two congeneric host plants—such as blueberry and cranberry. Mathur et al. [9] also revealed that—based on the mitochondrial *cytochrome c oxidase subunit I* (*COI*) sequence differences—cryptic speciation occurred between *D. oxycoccana* populations on cranberry and highbush blueberry. This study revealed a 10.7–13.1% divergence between cranberry and blueberry *D. oxycoccana* samples on *COI* sequences, whereas little divergence was observed within cranberry (0–1.2%) or blueberry (0–1.3%) sample sequences [9]. This genetic difference between the two *D. oxycoccana* populations from blueberry and cranberry seems relatively large when compared to the genetic difference between the two host plants (blueberry versus cranberry), which is ca. 2.7% based on six chloroplast and two mitochondrial SSR loci [64]. Earlier studies by Fitzpatrick et al. [16] further support HAD in *D. oxycoccana* by showing that populations from blueberry and cranberry produce and respond to different sex pheromones. Thus, based on these previous studies and our current study, it is apparent that cryptic speciation has likely occurred between populations of *D. oxycoccana* on blueberry and cranberry.

Early on, Walsh [65] proposed a scenario to understand the occurrence of sympatric HAD that could lead to ecological speciation. This scenario proposes that, by switching to new host plants, phytophagous insects exploit novel ecological niches that can result in genetic isolation, subsequently leading to speciation [66]. HAD has been shown to cause host shifts mainly in monophagous insects, i.e., individuals of a population switch to a new, related host and then adapt and evolve through ecological isolation to utilize this newly acquired host [66]. It is proposed that ecological specialization of host-associated populations can result in species diversification when individuals with greater fitness on one resource preferentially mate with other individuals on that same resource [67], which is often correlated with oviposition site selection [68,69]. Therefore, mating and oviposition on the same host plant can facilitate genetic differentiation and lead to reproductive isolation among insect populations [70,71]. Once formed, host-associated insect populations can be maintained by ecologically mediated reproductive isolation [72], which could potentially account for the origin of the species [66]. Moreover, certain characteristics within agricultural ecosystems may increase the probability for HAD to occur in insect herbivores [73]—they include relatively long-standing evolutionary relationships between herbivores and their host plants, which is likely the case for *D. oxycoccana* associations with blueberry and cranberry in their native ranges.

Interestingly, a previous study also revealed possible HAD between blueberry and cranberry populations of the cranberry fruitworm—*Acrobasis vaccinii* Riley—a frugivorous pest species native to North American that feeds on blueberries and cranberries [73]. However, although *A. vaccinii* exhibits allochronic isolation based on distinct phenologies of populations from blueberries and cranberries, genetic differences between host-related populations were not clearly detected in the population genetics analyses [73]. Thus, unlike the previous study [73], our results show a clear genetic differentiation between the two host-associated *D. oxycoccana* populations from blueberry and cranberry, which indicates that ecological speciation occurred on these host plants. Corroborating the previous studies [9,17,74], our study shows that *D. oxycoccana* from blueberry and cranberry should be considered as two distinct species. Future taxonomic studies are needed to describe *D. oxycoccana* from blueberry and cranberry as separate valid species.

### 4.2. Genetic Structure and Fragmentation within D. oxycoccana

Although *D. oxycoccana* populations from blueberries and cranberries separated from each other as indicated above, the STRUCTURE analysis of *K* = 2 showed that some blueberry populations appear to be genetically close to the cranberry populations. Among the blueberry subgroups, subgroup D (Figure 3 and Figure 4) was the closest to the cranberry populations. In particular, two blueberry populations (US-B-GA1 and KR-B-JJ2) possess an intermediate genetic signature that aligns between the two host-associated populations (Figure 4; Appendix A). The DIYABC test also inferred that the blueberry populations from Georgia were closely related to the cranberry populations (Figure 5). In our analysis, we did not consider the different species of cultivated blueberry or their variety (Appendix A); thus, future studies need to better explain the intermediate divergence process of *D. oxycoccana* between the two populations from blueberry and cranberry.

Within blueberry populations, five distinct subgroups were detected according to our PCoA and STRUCTURE analyses (Figure 3 and Figure 4). Although these groups were not separated geographically, they are genetically different, possibly due to some ecological isolation factors. In the United States, the New Jersey, Georgia, and Michigan populations are separated by considerable genetic distances from each other (Figure 3 and Figure 4), which appear to be associated with differences in blueberry species or varieties rather than with geographical isolation. Indeed, commercial blueberries are composed of multiple species and their interspecific hybrids from the *Vaccinium* section *Cyanococcus* A. Gray, including *V. angustifolium*, *V. corymbosum*, *V. virgatum*, and *V. darrowii* Camp [64]. Blueberries were first domesticated in New Jersey (USA) in 1908 [75], and since then several varieties have been developed particularly for larger fruit size, increased concentrations of phytochemicals for improved human health and flavor, broadened phenological adaptations such as reduced chilling requirements, and increased yield [76]. Therefore, in addition to separating populations associated with blueberry or cranberry, *D. oxycoccana* populations could be further separated based on blueberry species and variety, despite its relatively short domestication period. In addition, the possible existence of host races needs to be tested in future studies.

*Dasineura oxycoccana*—like other cecidomyiids—has a short adult lifespan (2–3 days) [4,77]; thus, it is not considered a migratory species. This information suggests that the diversity in the genetic structure of populations and geographic isolation can be mostly explained by the spread of infested host plant material by humans. The genetic diversity of *D. oxycoccana* populations can also be explained by differences in blueberry species or variety cultivated in a specific area. As in the case of HAD between cranberry and blueberry, our results suggest the possibility of ecological isolation based on their association with specific blueberry species and variety—due to crop features (i.e., chemistry) as well as phenology and/or geographical fragmentation due to differences in cultivation environments.

Generally, the genetic diversity of the newly colonized *D. oxycoccana* populations from Korea is expected to be lower than that of the populations from its native US range. However, according to the genetic diversity indices (*H*o, *H*s, *N*_A_, and *R*_S_) reported in Table 1, the Korean and USA populations seem to have similar levels of genetic diversity. This finding is consistent with the results that most of the populations did not suffer from genetic bottleneck in the BOTTLENECK test (Appendix A)—supporting the genetic diversity of *D. oxycoccana* settled in different Korean regions. However, further investigation is needed to determine whether this is closely related to the diversity of blueberry seedlings from imported routes or cultivar diversity in the area.

### 4.3. Inferring Introductions from Source to Invaded Regions

The recent increase in the import and export of goods due to international trade has likely caused the unintentional introductions of invasive insect pests [78,79]—as was the case of the *D. oxycoccana* invasion into Korea. In fact, there have been many study cases of trans-Pacific introductions by the unintentional transportation of exotic insects [34,80,81,82], such as the introduction of the imported fire ant—*Solenopsis invicta* (Buren)—from the United States to East Asia [34]. On the other hand, the soybean aphid—*Aphis glycines* Matsumura—originated from East Asia and spread to the eastern and central regions of the United States and was shown to exhibit low genetic variation and diversity in the invaded regions compared to its native range [80]. Other species that originated in Asia and invaded the United States include the Asian long-horned beetle—*Anoplophora glabripennis* (Motschulsky) [81]—and more recently the spotted lanternfly—*Lycorma delicatula* (White) [82]. In most of these cases, the exotic insects spread through infested host plants or soil into other regions, and there is a tendency to have multiple introductions due to the bridgehead effect, where the initial invasive populations serve as the source of additional invasions via secondary introductions [29,30,34,82]. Therefore, population genetics studies on invasive species are useful for inferring an introduction from the source to the invaded regions.

In this study, *D. oxycoccana* populations collected from the invaded Korea segregated into the subgroups A, B, D, and E (Figure 3 and Figure 4). In particular, because many individuals from Korea have similar dominant genotypes found in the subgroups A and B, it is likely that blueberry plants of a specific species or variety were introduced from the same source into Korea (Figure 3 and Figure 4). On the other hand, some populations such as KR-B-JJ2 (subgroup D) and KR-B-CA (subgroup E) were apparently close to US populations from Georgia and Michigan, respectively, based on the results from PCoA and STRUCTURE (Figure 3 and Figure 4). In addition, the average *F*_st_ value among the *D. oxycoccana* populations from Michigan, New Jersey, and Georgia was 0.136, whereas those between KR-B-JJ2 and Georgia populations or between KR-B-CA and Michigan populations were rather lower (0.129 and 0.036, respectively) (Appendix A), which suggests a highly regional association between the source and the invasive *D. oxycoccana* populations. 

Although the *D. oxycoccana* populations of subgroups A and B from Korea are most similar to those from New Jersey in the eastern United States, our data are insufficient to conclusively say that the genetic structure of the invasive populations in Korea is related to a specific US region. Nevertheless, based on the DIYABC analysis, it was determined that the possibility of the independent introductions of subgroups A and B was high, estimating a scenario that included an unsampled population (Figure 5), indicating that *D. oxycoccana* from Korea likely came from an unknown region (unsampled population)—in the United States or from a country in Europe or Asia—and that the invasion of subgroup B occurred earlier than that of subgroup A (Figure 5; Appendix A). These results suggest that *D. oxycoccana* populations in Korea spread by multiple introductions with genetic origins from at least four independent genotypes (subgroups A, B, D, and E) (Figure 4 and Figure 5), which strongly suggests that they were introduced by humans through imported blueberry nursery plants. In fact, even within North America, this species has likely been moved across regions with the transport of infested host-plant nursery plants [2,9]. Moreover, *D. oxycoccana* populations in the invaded Korea show a lack of geographically based genetic structure but are, however, genetically similar within each of the subgroups. In this study, samples from other countries such as China and Japan—which could be sources of invasion into Korea—were not available, so there are limitations that can be inferred from our analysis. Future genetic studies should expand the sampling of *D. oxycoccana* populations to include other countries in North America and Asia as well as Europe.

## 5. Conclusions

Our results demonstrate the following: (1) a clear separation between the two host-associated *D. oxycoccana* populations from blueberry and cranberry, which could be considered distinct species; (2) the occurrence of five genetically isolated *D. oxycoccana* subgroups from blueberry; (3) that multiple *D. oxycoccana* introductions (subgroups A, B, D, and E in Figure 4) likely occurred in Korea; and (4) that the dominant invasive *D. oxycoccana* population (subgroups A and B in Figure 4) from Korea was likely introduced from a genotypically diverse bridgehead population, possibly from an unsampled source population rather than directly from the pest’s native range. These findings will help to better identify *D. oxycoccana* populations associated with their host plant (i.e., blueberry or cranberry). They will also facilitate the identification of the source of *D. oxycoccana* into newly invaded regions, which may help prevent a bridgehead effect in these invaded regions.

## Figures and Tables

**Figure 1 insects-13-00880-f001:**
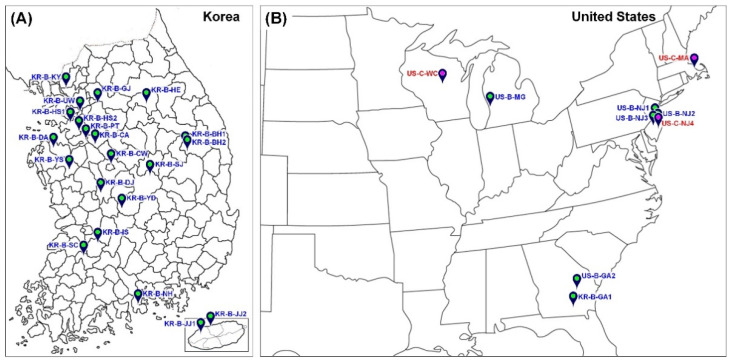
Sampling collection sites of *Dasineura oxycoccana.* (**A**) South Korea. (**B**) USA. Detailed information of locations is described in Appendix A Appendix A.

**Figure 2 insects-13-00880-f002:**
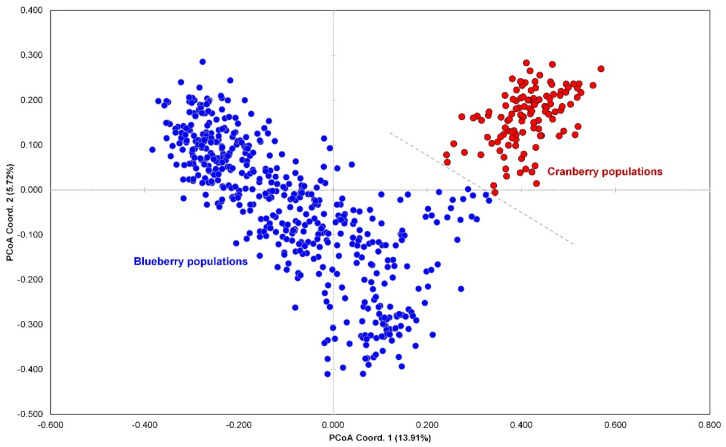
Principal coordinate analysis (PCoA) plotted using microsatellite data from 632 *Dasineura oxycoccana* individuals collected from 31 blueberry and cranberry populations from USA and Korea. The X-axis is coordinate 1, ranging from −0.60 to 1.80, and the Y-axis is coordinate 2, ranging from −0.50 to 0.40. Red circles correspond to cranberry populations, and blue circles correspond to blueberry populations.

**Figure 3 insects-13-00880-f003:**
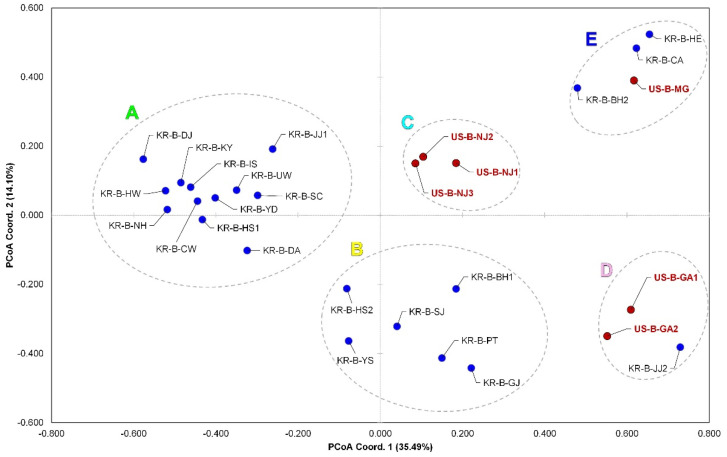
Principal coordinate analysis (PCoA) plotted using microsatellite data from 28 *Dasineura oxycoccana* blueberry populations from USA and Korea. The *X*-axis is coordinate 1, ranging from −0.80 to 0.80, and the *Y*-axis is coordinate 2, ranging from −0.60 to 0.60. Red circles correspond to the USA populations, and blue circles correspond to the Korean populations. Letters A–E represent genetically isolated subgroups.

**Figure 4 insects-13-00880-f004:**
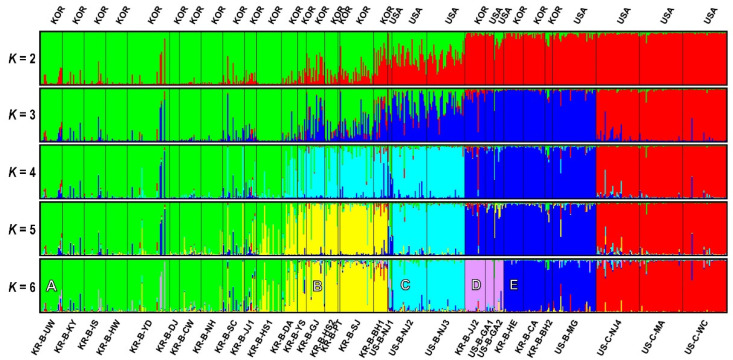
Bayesian clustering (STRUCTURE) for the 31 *Dasineura oxycoccana* populations collected from blueberry and cranberry in the USA and Korea. Individual assignment plots for *K* = 2, 3, 4, 5, and 6. Different colors indicate different clusters (e.g., A–E).

**Figure 5 insects-13-00880-f005:**
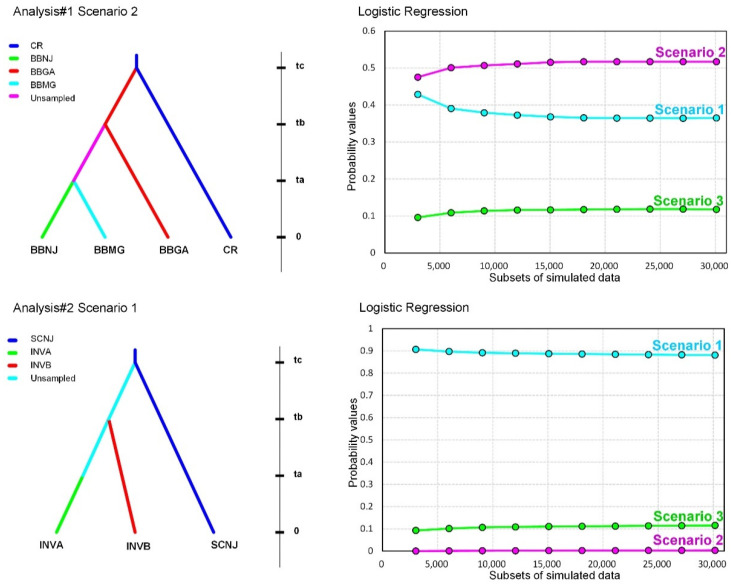
Scenarios (left panels) most supported by two independent analyses (Analysis #1 and Analysis #2), and probability values (right panels) from corresponding logistic regressions. Analysis #1 tested for serial divergence between *Dasineura oxycoccana* populations from blueberry and cranberry in the native range, whereas Analysis #2 tested for a divergence between a source (USA) of *D. oxycoccana* and the invaded region (Korea). Time (not to scale) is indicated on the right side of each scenario.

**Table 1 insects-13-00880-t001:** Summary statistics for microsatellite data from *Dasineura oxycoccana* populations.

Pop. ID	No.	*H*o (SD)	*H*e (SD)	HWE	*H* _S_	*N* _A_	*R* _S_	*F_IS_*
KR-B-UW	20	0.788 (0.051)	0.740 (0.024)	Ns	0.74	6.83	1.74	−0.07
KR-B-GJ	17	0.692 (0.068)	0.727 (0.024)	Ns	0.73	6.17	1.73	0.06
KR-B-HS1	23	0.803 (0.052)	0.698 (0.018)	Ns	0.70	6.75	1.70	−0.15
KR-B-HS2	12	0.788 (0.051)	0.740 (0.024)	Ns	0.70	6.69	1.72	−0.13
KR-B-KY	20	0.792 (0.059)	0.686 (0.026)	Ns	0.68	5.25	1.69	−0.16
KR-B-PT	2	0.708 (0.114)	0.653 (0.075)	Ns	0.68	2.42	1.65	−0.13
KR-B-HE	18	0.880 (0.032)	0.697 (0.024)	* excess	0.69	5.75	1.70	−0.27
KR-B-CW	20	0.817 (0.059)	0.706 (0.022)	Ns	0.70	6.00	1.71	−0.16
KR-B-YD	39	0.738 (0.052)	0.705 (0.017)	Ns	0.70	8.08	1.70	−0.05
KR-B-DA	15	0.769 (0.061)	0.699 (0.022)	Ns	0.70	5.25	1.70	−0.10
KR-B-CA	20	0.821 (0.042)	0.732 (0.023)	Ns	0.73	6.17	1.73	−0.13
KR-B-DJ	9	0.898 (0.040)	0.619 (0.027)	* excess	0.60	3.67	1.62	−0.49
KR-B-YS	8	0.891 (0.035)	0.719 (0.040)	Ns	0.68	4.50	1.72	−0.30
KR-B-IS	20	0.721 (0.064)	0.701 (0.026)	Ns	0.70	6.17	1.70	−0.03
KR-B-SC	20	0.809 (0.049)	0.721 (0.025)	Ns	0.72	6.67	1.72	−0.13
KR-B-HW	20	0.713 (0.072)	0.649 (0.033)	Ns	0.65	3.92	1.65	−0.10
KR-B-BH1	13	0.788 (0.062)	0.699 (0.051)	Ns	0.69	6.08	1.70	−0.14
KR-B-BH2	7	0.833 (0.060)	0.782 (0.026)	Ns	0.78	5.33	1.78	−0.07
KR-B-SJ	31	0.805 (0.049)	0.734 (0.029)	Ns	0.73	7.83	1.73	−0.10
KR-B-NH	20	0.825 (0.064)	0.647 (0.036)	* excess	0.64	4.83	1.65	−0.29
KR-B-JJ1	19	0.626 (0.081)	0.720 (0.061)	* deficit	0.72	7.08	1.72	0.13
KR-B-JJ2	11	0.659 (0.077)	0.714 (0.028)	Ns	0.72	5.58	1.71	0.08
US-B-GA1	8	0.762 (0.072)	0.729 (0.028)	Ns	0.73	4.58	1.73	−0.05
US-B-GA2	9	0.731 (0.059)	0.689 (0.038)	Ns	0.69	5.08	1.69	−0.07
US-B-NJ1	4	0.681 (0.069)	0.788 (0.054)	Ns	0.81	4.42	1.79	0.16
US-B-NJ2	32	0.784 (0.048)	0.779 (0.030)	Ns	0.78	11.17	1.78	−0.01
US-B-NJ3	35	0.848 (0.046)	0.789 (0.019)	Ns	0.79	9.92	1.79	−0.08
US-B-MG	40	0.759 (0.077)	0.776 (0.033)	* deficit	0.78	11.08	1.78	0.02
US-C-NJ4	40	0.697 (0.094)	0.681 (0.062)	Ns	0.68	10.83	1.68	−0.02
US-C-MA	40	0.655 (0.119)	0.596 (0.090)	Ns	0.60	8.17	1.60	−0.10
US-C-WC	40	0.617 (0.119)	0.538 (0.084)	Ns	0.54	7.42	1.54	−0.15

No. = number of individuals; *H*o = observed heterozygosity; *H*e = expected heterozygosity; HWE = Hardy–Weinberg Equilibrium; *H*_S_ = gene diversity; *N*_A_ = mean number of alleles; *R*_S_ = allelic richness; *F*_IS_ = estimates of inbreeding coefficient (Weir and Cockerham (1984)); SD = standard deviation; Ns = non-significance (*p* > 0.05); * = significant differences for heterozygote excess or deficit (*p* < 0.0001). In the population identifiers (Pop. ID), ‘-B-‘ means a collection from blueberry, while ‘-C-‘ means a collection from cranberry.

**Table 2 insects-13-00880-t002:** Analysis of molecular variance (AMOVA) of 632 individuals in 31 populations of *Dasineura oxycoccana* in Korea and USA. Case 1: cranberry-associated versus blueberry-associated. Case 2: source (USA) versus invasive (KOR). Case 3: genetic structure-based groups (A, B, C, D, E).

Case	Among Groups			Among Populations within Groups			Within Populations		
	V_a_	PV	*p*	V_b_	PV	*p*	V_c_	PV	*p*
1	0.82	14.58	<0.0001	0.59	10.62	<0.0001	4.18	74.80	<0.0001
2 *	0.38	7.49	<0.0001	0.42	8.34	<0.0001	4.31	84.17	<0.0001
3 *	0.54	10.65	<0.0001	0.22	4.32	<0.0001	4.31	85.03	<0.0001

* Only *D. oxycoccana* populations from blueberry were included in the analysis.

## Data Availability

Data are available upon request from the authors.

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
