# Peer review of "Population Genetics of the Blueberry Gall Midge, Dasineura oxycoccana (Diptera: Cecidomyiidae), on Blueberry and Cranberry and Testing Invasion Scenarios"

_insects, 2022, doi:10.3390/insects13100880_

Round 1

Reviewer 1 Report

The document reports a population genetics study with Dasineura oxycoccana, a phytophagous dipteran that is considered a pest of blueberry and cranberry in the native geographical area of North America, and has expanded its distribution to Europe and some Asian countries. Based on the patterns of genetic diversity and levels of differentiation obtained with 12 nuclear molecular markers in 31 populations (22 from Korea and 9 from the US), the authors apparently intend to support a host plant-associated speciation process of D. oxycoccana and the origin of the populations sampled in Korea. The study reveals high levels of genetic diversity in the populations of D. oxycoccana and a relatively high level of differentiation between host-associated populations (14%), so the authors propose to assume the formation of two species. Patterns of genetic similarity are inconclusive regarding the origin of the Korea populations; however, it was observed that the populations were possibly formed from individuals originating from NJ.

The information reported is undoubtedly very valuable, and the data have been analyzed with modern and sophisticated methods. However, I believe that the presentation needs to be improved to clarify the points that are of interest to the authors and to make it easier to follow their argument. The study contemplates the analysis of a large batch of data and covers a wide geographical area in Korea, while for the region of origin of D. oxycoccana (US and Canada) it is geographically smaller. Although the authors highlight that their study has a larger number of data than previous works, I think it is necessary to recognize that it is still a work with unbalanced sample sizes, to the extent that one of their invasion scenarios indicates that the origin may be in an unsampled population.

It is important to incorporate as a research objective the identification of the process of D. oxycoccana species formation by the differentiated use of host species, and to support this objective in the introduction. The second objective is better stated and is consistent with the information in the introduction, methods and discussion, but with respect to the two-species proposal, it seems to me that the authors' statements in the introduction are not entirely clear. The statistical analyses are generally correct, but it would be necessary to point out which patterns of diversity estimators and genetic differentiation are to be expected if we are facing a speciation process. In the discussion there is ample argumentation in favor of speciation, so it is evident that it is an objective of the authors. It seems to me that it is necessary to make this objective explicit, and to offer more arguments in this respect in the introduction. In the methodology, it is necessary to highlight which analyses will allow inferences to be made regarding speciation. The intention to compare host-associated populations of D. oxycoccana is intended to show speciation, so it should be pointed out in that way (and of course the respective arguments should be given).

Another aspect I suggest reviewing is the writing style. It was not possible for me to have a fluent reading and keep my interest. Also, I believe that a restructuring of the information would be convenient to improve the comprehension of the study. The need to read a paragraph more than once to understand its content is distracting to grasp the important message of this document.

The following are some specific observations that I hope will be useful:

13-17. It may be appropriate to reduce this information further, and to foreground the interest in confirming the speciation of D. oxycoccana promoted by the use of different host species.

19-20. Add some benefit to be gained by successfully mapping invasion routes.

42-62. It may possibly help the reader to separate the arguments that are directed to the processes of distribution extension and those that have to do with speciation by differentiation of populations due to the host plant.

63-64. How are the biotypes distinguished? The idea in the sentences do not connect well.

63-74. This information could be reorganized so that the hypothesis of develop of host specificity remains at the forefront of the information.

75-91. This paragraph also mentions its economic importance and the widening of its distribution. Then I perceive that it may be better to integrate the information by topics. The sequence of topics could be (1) D. oxycoccana as a pest, (2) its changes in distribution, (3) the importance of tracing invasion routes for the management of host plants or of the diptera itself, (4) then the formation of races by the use of host plants, which can be reinforced by the changes in its distribution due to the management of its host plants.

102-103. Consider rephrasing your objectives, and in particular I see that species formation is a specific interest of the authors, and I think it should come to the forefront.

116. Add a section with general aspects of the biology of D. oxycoccana. Life span, pupation sites, courtship aspects, selection of species for oviposition. This information is useful to reinforce the speciation proposal by the use of different species.

172-173. Did you identify errors? If yes, at what rate and what was done? Was there stuttering? If so, how was it treated?

172. I think it would enhance the understanding of the information regarding the speciation process you wish to support to say what results are expected as evidence of speciation.

273-274. "likely due to...genotypes" bring the idea to the discussion.

283. I suggest including regional averages, and in the case of the US, those corresponding to each host-associated population.

313-320. It is not clear what the significance of showing a bottleneck is.

412-413. The first sentence should go as an objective in the introduction section.

416-417. In my opinion, the result is evidence, plus ecological speciation. Perhaps another sampling design is needed for it to constitute corroboration of the speciation process.

419. Perhaps the sequence of the argument is how the observed complete assortative mating impacts the diversity and structure of host-associated populations and thus corroborates or confirms ecological speciation with the genetic data of the present study. The genetic difference with micros is in agreement with the COI results, inferring that the host-associated populations are in an advanced stage of ecological speciation.

440-456. The information in this section could be moved to the introduction, and retaking the hypothesis would be consistent with the results.

467-469. This is a conclusion more in line with their results.

470. This section could go first, as it discusses genetic diversity and distribution within and between populations, and some of its causes. Then we discuss (in the corresponding section) whether these patterns of differentiation correspond to a process of ecological speciation.

510-525. This information seems more introduction than discussion. I suggest that it be rephrased to discuss the result more.

527-530. Consider reviewing, as they appear to be results rather than discussion.

543-544, It is a contradictory idea to line 530.

557. No discussion of bottleneck results was made.

561. I suggest that a possible probable cause, of the structuring, be offered.

Author Response

Reviewer #1

The document reports a population genetics study with Dasineura oxycoccana, a phytophagous dipteran that is considered a pest of blueberry and cranberry in the native geographical area of North America, and has expanded its distribution to Europe and some Asian countries. Based on the patterns of genetic diversity and levels of differentiation obtained with 12 nuclear molecular markers in 31 populations (22 from Korea and 9 from the US), the authors apparently intend to support a host plant-associated speciation process of D. oxycoccana and the origin of the populations sampled in Korea. The study reveals high levels of genetic diversity in the populations of D. oxycoccana and a relatively high level of differentiation between host-associated populations (14%), so the authors propose to assume the formation of two species. Patterns of genetic similarity are inconclusive regarding the origin of the Korea populations; however, it was observed that the populations were possibly formed from individuals originating from NJ.

The information reported is undoubtedly very valuable, and the data have been analyzed with modern and sophisticated methods. However, I believe that the presentation needs to be improved to clarify the points that are of interest to the authors and to make it easier to follow their argument. The study contemplates the analysis of a large batch of data and covers a wide geographical area in Korea, while for the region of origin of D. oxycoccana (US and Canada) it is geographically smaller. Although the authors highlight that their study has a larger number of data than previous works, I think it is necessary to recognize that it is still a work with unbalanced sample sizes, to the extent that one of their invasion scenarios indicates that the origin may be in an unsampled population.

>> [Response] There are many practical restrictions on the sampling of D. oxycoccana. We acknowledge that sampling limited to select regions in the USA has limitations. However, research was conducted to derive the most objective results within a limited sampling. Thank you for your sincere comments.

It is important to incorporate as a research objective the identification of the process of D. oxycoccana species formation by the differentiated use of host species, and to support this objective in the introduction. The second objective is better stated and is consistent with the information in the introduction, methods and discussion, but with respect to the two-species proposal, it seems to me that the authors' statements in the introduction are not entirely clear. The statistical analyses are generally correct, but it would be necessary to point out which patterns of diversity estimators and genetic differentiation are to be expected if we are facing a speciation process. In the discussion there is ample argumentation in favor of speciation, so it is evident that it is an objective of the authors. It seems to me that it is necessary to make this objective explicit, and to offer more arguments in this respect in the introduction. In the methodology, it is necessary to highlight which analyses will allow inferences to be made regarding speciation. The intention to compare host-associated populations of D. oxycoccana is intended to show speciation, so it should be pointed out in that way (and of course the respective arguments should be given).

>> [Response] There have been several previous studies on the host-associated speciation of D. oxycoccana. An area of interest in our study is whether they have achieved sufficient genetic isolation on the nuclear genome. To prove this, we had developed nuclear markers by isolating microsatille loci from D. oxycoccana in a previous study (Kim et al. 2015), and applied it as a host-associated population of D. oxycoccana in this study. Therefore, as a result, the blueberry-associated and cranberry-associated populations of D. oxycoccana with more than 600 individuals demonstrated that at least the allelic pattern on multiloci was completely different and did not share, that is, genetic isolation certainly exists.

Another aspect I suggest reviewing is the writing style. It was not possible for me to have a fluent reading and keep my interest. Also, I believe that a restructuring of the information would be convenient to improve the comprehension of the study. The need to read a paragraph more than once to understand its content is distracting to grasp the important message of this document.

The following are some specific observations that I hope will be useful:

13-17. It may be appropriate to reduce this information further, and to foreground the interest in confirming the speciation of D. oxycoccana promoted by the use of different host species.

>> agreed. “Native to Central and Eastern North America,” was removed. The meaning of the use of different host species is expressed to “..insect pest of blueberry and cranberry in its native range..”. in lines 13-16.

19-20. Add some benefit to be gained by successfully mapping invasion routes.

>> agreed. We mentioned the benefit in lines 92-94 because of quantity limit of abstract.

42-62. It may possibly help the reader to separate the arguments that are directed to the processes of distribution extension and those that have to do with speciation by differentiation of populations due to the host plant.

>> Thank you for your valuable comments. We have modified the relevant content accordingly.

63-64. How are the biotypes distinguished? The idea in the sentences do not connect well.

>> deleted the sentence. Meaning of biotypes is wrong in this study. Thank you.

63-74. This information could be reorganized so that the hypothesis of develop of host specificity remains at the forefront of the information.

>> [Response] As a result of re-examination, there was no major problem in information transmission of the hypothesis of develop of host specificity, so it was left as it is.

75-91. This paragraph also mentions its economic importance and the widening of its distribution. Then I perceive that it may be better to integrate the information by topics. The sequence of topics could be (1) D. oxycoccana as a pest, (2) its changes in distribution, (3) the importance of tracing invasion routes for the management of host plants or of the diptera itself, (4) then the formation of races by the use of host plants, which can be reinforced by the changes in its distribution due to the management of its host plants.

>> As you pointed out, I changed the order of the paragraphs in the introduction. However, the order we think of is (1) D. oxycoccana as a pest, (2) the formation of races by the use of host plants, (3) its changes in distribution, (4) the importance of tracing invasion routes for the management of host plants or of the diptera itself. Since the part that is proven in the study comes first, the order of the introduction is also matched accordingly. (lines 43-113)

102-103. Consider rephrasing your objectives, and in particular I see that species formation is a specific interest of the authors, and I think it should come to the forefront.

>> [Response] The formation of races by the use of host is the first objective of the purpose. So that is okay in order of appearance for objectives as “The main objectives of this study were to compare the population genetic structure of D. oxycoccana between 1) populations collected from cranberries and blueberries; and 2) a region of origin, i.e., USA, and an invaded region, i.e., Korea. Specifically, we first compared D. oxycoccana samples collected from blueberries and cranberries to test for HAD in this species.”

  1. Add a section with general aspects of the biology of D. oxycoccana. Life span, pupation sites, courtship aspects, selection of species for oviposition. This information is useful to reinforce the speciation proposal by the use of different species.

>> [Response] Life span is mentioned in Line 535. Other ecological characteristics can be found in the first paragraph (Lines 51-63) and also in the related references [4, 77].

172-173. Did you identify errors? If yes, at what rate and what was done? Was there stuttering? If so, how was it treated?

>> [Response] I tested those but any errors (large-allele dropout and stuttering) were not found.

  1. I think it would enhance the understanding of the information regarding the speciation process you wish to support to say what results are expected as evidence of speciation.

>> [Response] MICROCHECKER was used to check the null allele or stuttered peaks. It is not considered to detect the speciation process between host-associated populations.

273-274. "likely due to...genotypes" bring the idea to the discussion.

>> corrected. It was converted from "likely due to” to “which was affected by”. (line 271)

  1. I suggest including regional averages, and in the case of the US, those corresponding to each host-associated population.

>> [Response] Those were demonstrated in lines 325-334.

313-320. It is not clear what the significance of showing a bottleneck is.

>> [Response] As mentioned in the text, the population KR-B-HW has apparently undergone genetic bottleneck since it was significance in all analyses, but the bottleneck test should be cautiously interpreted because the sample size for some populations was less than 30 individuals because the sampled individual number of KR-B-HW are 20. (lines 315-322)

412-413. The first sentence should go as an objective in the introduction section.

>> corrected it to “Regarding possible ecological speciation between the two races of D. oxycoccana from blueberry and cranberry,” (lines 414-415)

416-417. In my opinion, the result is evidence, plus ecological speciation. Perhaps another sampling design is needed for it to constitute corroboration of the speciation process.

>> [Response and rebuttal] There have been several previous studies on the host-associated speciation of D. oxycoccana. An area of interest in our study is whether they have achieved sufficient genetic isolation on the nuclear genome. To prove this, we had developed markers by isolating microsatille loci from D. oxycoccana in a previous study, and applied it as a host-associated population of D. oxycoccana in this study. Therefore, as a result, the blueberry-associated and cranberry-associated populations of D. oxycoccana with more than 600 individuals demonstrated that at least the allelic pattern on multiloci was completely different and did not share, that is, genetic isolation certainly exists. Some study said that, with less than 30 samples, the speciation occurred between them.

  1. Perhaps the sequence of the argument is how the observed complete assortative mating impacts the diversity and structure of host-associated populations and thus corroborates or confirms ecological speciation with the genetic data of the present study. The genetic difference with micros is in agreement with the COI results, inferring that the host-associated populations are in an advanced stage of ecological speciation.

>> Thank you for your valuable comments. As you mentioned, our results based on the nuclear markers of microsatellite are consistent with the previous results based on the haplotypes of COI.

440-456. The information in this section could be moved to the introduction, and retaking the hypothesis would be consistent with the results.

>> [Response and rebuttal] Since it's focused on ecological speciation, placing it within a discussion fits the contextual flow. It contains very heterogeneous content to be described as an introduction.

467-469. This is a conclusion more in line with their results.

>> corrected “we conclude” as “our study shows”. (line 469)

  1. This section could go first, as it discusses genetic diversity and distribution within and between populations, and some of its causes. Then we discuss (in the corresponding section) whether these patterns of differentiation correspond to a process of ecological speciation.

>> [Response and rebuttal] This subheading is the second sub-topic because fragmented structure is mostly applicable only to blueberry population.

510-525. This information seems more introduction than discussion. I suggest that it be rephrased to discuss the result more.

>> [Response and rebuttal] Since it's more deeply focused on Inferring introductions from source to invaded regions, placing it within a discussion fits the contextual flow. It contains very heterogeneous content to be described as an introduction.

527-530. Consider reviewing, as they appear to be results rather than discussion.

>> [Response and rebuttal] This is the discussion about the interpretation of results that are not covered in Results. I don't think it should be moved to results.

543-544, It is a contradictory idea to line 530.

>> corrected “the same US source” to “the same source”. As we used unsampled source in ABC analysis, the source cannot be clearly determined to US site. Thank you (line 542)

  1. No discussion of bottleneck results was made.

>> [Response and rebuttal] As mentioned above, the population KR-B-HW has apparently undergone genetic bottleneck since it was significance in all analyses, but the bottleneck test should be cautiously interpreted because the sample size for some populations was less than 30 individuals because the sampled individual number of KR-B-HW are 20. Therefore, no significant bottleneck was found in this study. So as you know, there is nothing to discuss. (lines 315-322)

  1. I suggest that a possible probable cause, of the structuring, be offered.

>> [Response and rebuttal] As you know, only conclusions are given here. All probable causes have already been presented in the discussion. Redeploying the duplicate content is meaningless and therefore not described. Please read the contents of the discussion.

Reviewer 2 Report

This research provide an interesting results on the genetic structure of the blueberry gall midage Dasineura oxycocana, an invasive species in Korea. The analysis will be helpful to understand the source of this pest in Korea. I think the manuscript was well written. It can be accepted after minor revision.

1) The result in Abstract and the conclusion is not clear. According to the genetics structure (Bayesian clustering), the source of this pest in Korea seems to be multiple. However, the coclusion in the paper is "invasive D. oxycoccana population from Korea was likely introduced from a genotypically diverse bridgehead population, which was likely introduced from an unsampled source population rather than directly from native range."

2) In the manuscript, "Delta K values calculated by Evanno et al. (2005) method detecting K = 2 groups". Actually, it has been revealed that the K will be 2 for the invasive species because of the potential effects of introductions. Thus, the potential effects should be discussed in the section of "Discussion" and the reference should be cited in the manuscript.

Author Response

Reviewer #2

This research provide an interesting results on the genetic structure of the blueberry gall midage Dasineura oxycocana, an invasive species in Korea. The analysis will be helpful to understand the source of this pest in Korea. I think the manuscript was well written. It can be accepted after minor revision.

1) The result in Abstract and the conclusion is not clear. According to the genetics structure (Bayesian clustering), the source of this pest in Korea seems to be multiple. However, the coclusion in the paper is "invasive D. oxycoccana population from Korea was likely introduced from a genotypically diverse bridgehead population, which was likely introduced from an unsampled source population rather than directly from native range."

>> agreed. These were corrected as “3) multiple invasion of D. oxycoccana (subgroups A, B, D and E in Figure 4) at least four times occurred in Korea; 4) the dominant invasive D. oxycoccana population (subgroups A and B in Figure 4) from Korea likely introduced from a genotypically diverse bridgehead population; 5) which was likely introduced from an unsampled source population rather than directly from native range.” In conclusions (lines 575-582) and “3) multiple invasion of D. oxycoccana at least four times occurred in Korea; 4) the dominant invasive D. oxycoccana population from Korea likely introduced from a genotypically diverse bridgehead population; 5) which was likely introduced from an unsampled source population rather than directly from native range.” in abstract. (lines 22-26)

2) In the manuscript, "Delta K values calculated by Evanno et al. (2005) method detecting K = 2 groups". Actually, it has been revealed that the K will be 2 for the invasive species because of the potential effects of introductions. Thus, the potential effects should be discussed in the section of "Discussion" and the reference should be cited in the manuscript.

>> see lines 353-354, “Although K = 2 provided the best estimate, the results from K = 3 to K = 6 were further considered because they most effectively displayed the relationships among D. oxycoccana populations when compared with other analyses such as PCoA (Figure 3).” Since we mentioned this part in the Results section, we did not mention it repeatedly in a separate discussion. Actually, when K = 2, structure result showed just separation between cranberry and blueberry populations. Even although Evanno method determined K = 2 was the best likelihood, we demonstrated the results from K = 3 to K = 6 because they most effectively displayed the relationships among D. oxycoccana populations only on ‘BLUEBERRY’. Also, since interpreting the Evanno method is quite technical, cross-analysis with other analyzes is required such as PCoA (mentioned as above).

Reviewer 3 Report

Authors have prepared an excellent manuscript describing genetic diversity, genetic structure, and possible invasion route of blueberry gall midge, Dasineaura oxycoccana into Korea. I have a few comments and suggestions for further improvement of the manuscript.

1. Lines 99 – 100, it is mentioned here that China and Japan are also possible sources of D. oxycoccana invading Korea. However, the author did not include specimens from these countries in this study. Thus, it will be useful to state this limitation or provide an explanation.

2. Section 3.1; Generally, the genetic diversity of the newly colonized populations (Korea) is expected to be lower than the native range (USA). However, according to the genetic diversity indices (Ho, Hs, NA, RS) reported in Table 1, Korean and USA populations seem to have similar levels of genetic diversity. Therefore, it will be useful to provide a discussion on this topic.

3. Table 1, It will be useful to add additional information such as host plants in this Table. There are more details provided in Table S1 but it will be easier for the reader if it presents in this Table.

4. References in the reference list are not in the same format as the journal names, some are written in full (e.g. 1) but others are in abbreviation (e.g. 2). Therefore, please check and make their consistency.

Author Response

Reviewer #3

Authors have prepared an excellent manuscript describing genetic diversity, genetic structure, and possible invasion route of blueberry gall midge, Dasineaura oxycoccana into Korea. I have a few comments and suggestions for further improvement of the manuscript.

  1. Lines 99 – 100, it is mentioned here that China and Japan are also possible sources of D. oxycoccana invading Korea. However, the author did not include specimens from these countries in this study. Thus, it will be useful to state this limitation or provide an explanation.

>> agreed. We inserted related statement as “Unfortunately, in this study, samples from China and Japan, which were likely sources of invasion into Korea, were not available, so there are limitations that can be verified by analysis. Instead, the introduction scenario was estimated by including the unsampled population as an unknown source in the ABC analysis.” in 118-122.

  1. Section 3.1; Generally, the genetic diversity of the newly colonized populations (Korea) is expected to be lower than the native range (USA). However, according to the genetic diversity indices (Ho, Hs, NA, RS) reported in Table 1, Korean and USA populations seem to have similar levels of genetic diversity. Therefore, it will be useful to provide a discussion on this topic.

>> agreed. We inserted related statement as “Generally, the genetic diversity of the newly colonized populations from Korea is expected to be lower than the native range from USA. However, according to the genetic diversity indices (Ho, Hs, NA and RS) reported in Table 1, Korean and USA populations seem to have similar levels of genetic diversity. This is consistent with the results that most of the populations did not suffer from the genetic bottleneck in the BOTTLENECK test (Supplementary Material Table S3), supporting the genetic diversity of D. oxycoccana settled in different Korean regions. However, further investigation is needed to determine whether this is closely related to the diversity of seedling import routes or cultivar diversity in the area.”in 519-527.

  1. Table 1, It will be useful to add additional information such as host plants in this Table. There are more details provided in Table S1 but it will be easier for the reader if it presents in this Table.

 >> agreed. We inserted related statement as “In population identifiers, ‘-B-‘ means collection from blueberry, while ‘-C-‘ means collection from cranberry.” in footnote of Table 1.

  1. References in the reference list are not in the same format as the journal names, some are written in full (e.g. 1) but others are in abbreviation (e.g. 2). Therefore, please check and make their consistency.

>> This will be corrected in process of production. Thank you.

Round 2

Reviewer 1 Report

I appreciate that you have taken into account all my suggestions and that some have been incorporated in this version, as well as your kind response for those that were not included.

Reviewer 2 Report

I think the authors have explain the questions raised.